# Diagnosis of Soybean Diseases Caused by Fungal and Oomycete Pathogens: Existing Methods and New Developments

**DOI:** 10.3390/jof9050587

**Published:** 2023-05-18

**Authors:** Behnoush Hosseini, Ralf Thomas Voegele, Tobias Immanuel Link

**Affiliations:** Department of Phytopathology, Institute of Phytomedicine, Faculty of Agricultural Sciences, University of Hohenheim, Otto-Sander-Str. 5, 70599 Stuttgart, Germany; behnoush.hosseini@uni-hohenheim.de (B.H.); ralf.voegele@uni-hohenheim.de (R.T.V.)

**Keywords:** Soybean (*Glycine max*), seed-borne fungi, soil borne pathogens, molecular detection methods, PCR, qPCR, LAMP

## Abstract

Soybean (*Glycine max*) acreage is increasing dramatically, together with the use of soybean as a source of vegetable protein and oil. However, soybean production is affected by several diseases, especially diseases caused by fungal seed-borne pathogens. As infected seeds often appear symptomless, diagnosis by applying accurate detection techniques is essential to prevent propagation of pathogens. Seed incubation on culture media is the traditional method to detect such pathogens. This method is simple, but fungi have to develop axenically and expert mycologists are required for species identification. Even experts may not be able to provide reliable type level identification because of close similarities between species. Other pathogens are soil-borne. Here, traditional methods for detection and identification pose even greater problems. Recently, molecular methods, based on analyzing DNA, have been developed for sensitive and specific identification. Here, we provide an overview of available molecular assays to identify species of the genera *Diaporthe*, *Sclerotinia*, *Colletotrichum*, *Fusarium*, *Cercospora*, *Septoria*, *Macrophomina*, *Phialophora*, *Rhizoctonia*, *Phakopsora*, *Phytophthora*, and *Pythium,* causing soybean diseases. We also describe the basic steps in establishing PCR-based detection methods, and we discuss potentials and challenges in using such assays.

## 1. Introduction

Soybean (*Glycine max*) is among the most important crops. Soybean was domesticated in China over 3000 years ago and introduced to other Asian countries and, later, the Americas, Africa, and Europe [1]. Soybean production amounted to 355,605 million tons in 2021–2022 [2], which illustrates the enormous economic importance of this crop. Soybean is threatened by several abiotic and biotic stress factors, which result in reduction of soybean yield and quality [3]. Pathogens, pests, and weeds cause significant losses to soybean. In the first edition of “The Compendium of Soybean Diseases and Pests” [4], only 50 diseases were listed, while in the fifth edition more than 300 diseases are mentioned, of which 35 are classified as highly important [1]. The increase in disease incidence could at least partly be due to the intensification of soybean cultivation. Continuous growth or short crop rotations are favorable to several pathogens, which increase in density when the host plant is constantly available. With less intensive cultivation, these pathogens were less problematic than they are now. To limit economic losses, several measures are required, including cultural measures, seed treatment, efficient diagnostics, pesticides, and resistant cultivars [5].

Fungi in soybean seeds can cause a reduction in germination and establishment of seedlings, root diseases, and damping-off. Moreover, foliage and pod diseases are caused by fungal pathogens that considerably affect seed quality and quantity [1,6]. Fungi, together with a couple of oomycetes, are the most important soybean pathogens. The most important soybean fungal and oomycete pathogens are listed in Table 1 and illustrated in Figure 1.

The importance of these diseases varies over time. Environmental conditions and the susceptibility of cultivars are the main conditions defining the occurrence and dissemination of particular pathogens.

Soybean seed borne pathogens are carried by seeds as dormant mycelium, conidia/spores on the seed surface, or sclerotia mixed with seeds. Therefore, diagnosis of pathogens in seeds can reduce problems caused by these diseases. Detection and identification of fungal pathogens in seeds is still based on conventional methods, such as incubation of seeds on sterile filter paper or on semi-selective culture medium. Methods based on Polymerase Chain Reaction (PCR) techniques have been developed for the detection of several pathogens. Depending on how recently they were established, how extensively they have been tested, and to which purpose they were optimized, these assays can be highly specific, distinguishing species or even races of a given species. They allow identification of pathogens directly from infected tissue and sometimes even from soil without any cultivation, which makes them very fast. Methods, such as qPCR, also allow for a quantification of pathogens.

In the main part of this review we strive to include assays for diagnosis of most important fungal soybean pathogens. We will shortly introduce the pathogens, describe the methods and recount their history, and list the primers used. We also try to give information on the specificity of the different assays. In two additional sections we will describe the general steps of establishing PCR assays for diagnosis and further discuss their potential and difficulties both in establishing and in using these assays. Altogether, we hope that this will make our review a valuable resource for plant pathologists involved in the diagnosis of fungal soybean pathogens.

## 2. Common Fungal and Oomycete Soybean Pathogens and Molecular Assays to Detect Them

This part of the review is dedicated to important fungal and oomycete pathogens on soybean. The pathogens are organized by genus. While for some genera such as *Sclerotinia* or *Phakopsora,* only one or few species are economically important on soybean, the genus *Diaporthe* is foremost because of the number of species that can infect soybean. Here, we provide short descriptions of the species/genera. Since this is connected to molecular diagnosis, we provide some information on phylogenetic resolution of the genera and older and recent molecular assays.

We also list primers and, where possible, provide information on the level of specificity they provide for their targets. (The primer tables at the end of every subsection are not always mentioned in the text.) The melting temperatures for the primers (T_m_) given in the tables represent the temperatures actually used in the corresponding PCR programs, except where otherwise mentioned. Amplicon sizes are provided where available.

### 2.1. Genus Diaporthe

The genus *Diaporthe* Nitschke (1870) (asexual state *Phomopsis* (Sacc.) Bubák) includes hundreds of species. *Diaporthe* species are widespread and they are non-pathogenic endophytes (biotrophic fungi), saprotrophs, and fungal pathogens of many plants and even mammals [9,10].

Phytopathogenic *Diaporthe* species have been intensively studied, especially those affecting economically important crops, such as soybean, sunflower, grapes, citrus, and fruit, and ornamental trees [11]. Diseases caused by *Diaporthe* spp. on soybean are stem canker, pod and stem blight, and seed decay (Table 1) [7,12,13].

Morphological evaluation of fungal growth from surface sterilized soybean seeds plated on acidified potato dextrose agar (APDA) is still common in the identification of *Diaporthe* spp. [14]. However, because of strong similarities and overlaps in shapes and colors of cultures and in conidial size, delimitation of *Diaporthe* spp. is not valid just based on morphology [8,9,10]. Species diversity in the genus *Diaporthe* was explored by assays using PCR [8,15,16]. The nuclear ribosomal internal transcribed spacer (ITS) can be used for discrimination of *Diaporthe* spp. [17,18,19]. The primers Phom.I and Phom.II were designed on the ITS sequences of *D. phaseolorum* and *D. longicolla* for the detection of many *Diaporthe* spp. [15]. The primers DphLe and DphRi were developed for detection of *D. aspalathi* [20]. There are also real-time (q)PCR assays based on ITS to detect and quantify *Diaporthe* spp. on soybean. The first was developed by Zhang et al. [21]. The TaqMan primer-probe sets PL-3, PL-5, and DPC-3 were designed for *D. longicolla*, *D. aspalathi*, *D. caulivora*, and *D. sojae*. For additional information on the mentioned primers and assays, see Table 2.

However, ITS sequence data alone are not sufficient to resolve all *Diaporthe* species [23,24]. Therefore, translation elongation factor 1-α (*TEF1*), beta-tubulin (*TUB*), calmoduline (*CAL*), histone-3 (*HIS*), and large ribosomal subunit (*LSU*) are also used to differentiate *Diaporthe* species [25,26,27].

Recently, the TaqMan primer-probe sets DPCL, DPCC, DPCE, and DPCN were designed based on *TEF1* to identify, discriminate, and quantify *D. longicolla*, *D. caulivora*, *D. eres*, and *D. novem* simultaneously in a quadruplex real-time PCR [22].

In a study on expression of soybean defense-related genes, the *TUB*-based primers DPM were used with SYBR^®^ Green real-time PCR to detect and quantify *D. aspalathi* in soybean tissues [28]. The same assay was also used to quantify the fungal biomass in soybean stems infected with *D. caulivora* [29].

There are additional PCR assays for the identification of *Diaporthe* spp.: random amplified polymorphic DNA (RAPD), PCR-restriction fragment length polymorphism (RFLP), and amplified fragment length polymorphism (AFLP) have been used to distinguish *Diaporthe* species [15,16,30,31,32]. The species *D. caulivora*, *D. aspalathi,* and *D. sojae* could only be defined after finding differences between the *D. phaseolorum* varieties *caulivora*, *meridionalis*, and *sojae* using RAPD [30]. After that, PCR-RFLP was used to distinguish *D. longicolla*, *D. caulivora*, *D. aspalathi*, and *D. sojae* [15].

### 2.2. Genus Sclerotinia

*Sclerotinia sclerotiorum* (Lib.) de Bary is among the most destructive plant pathogens. It is the only relevant pathogen of the genus *Sclerotinia* on soybean. This fungus can infect over 400 species of plants, including sunflower, soybean, and oilseed rape [33]. On soybean *S. sclerotiorum* causes white mold, which reduces yield by more than 40% in wet and mild weather [34]. Since the pathogen disseminates via seeds, sowing seeds with certified health quality is the first step to avoid this disease.

Molecular detection has been established independently more than once. Primers SSFWD/SSREV were designed for the identification of *S. sclerotiorum* [35] and the specificity and sensitivity of these primers were confirmed experimentally by performing PCR and real-time (q)PCR (SYBR Green) using the touchdown method, to distinguish *S. sclerotiorum* isolates from *Aspergillus*, *Cercospora*, *Colletotrichum*, *Corynespora*, *Fusarium*, *Macrophomina*, *Diaporthe*, and *Botrytis* species [36]. Other primers were used to identify *S. sclerotiorum* ascospores and detect *S. sclerotiorum* in infected plant tissue [37,38,39]. When tested on soybean seeds inoculated with *S. sclerotiorum,* these primers were not useful [36]. Variability among isolates from this species from different regions or hosts can lead to these differences. In a separate study, the primer pair SSFWD/SSREV [35] was tested to detect *S. sclerotiorum* in inoculated and in naturally infected soybean seeds using the seed soaking procedure [34]. Since the ITS region did not allow for fully reliable diagnosis the mitochondrial small rRNA and SS1G_00263, coding for a hypothetical secreted protein was used [39,40]. For additional information on the mentioned primers and assays, see Table 3.

### 2.3. Genus Colletotrichum

The ascomycete genus *Colletotrichum* is quite large, with more than 200 species. Some of the species are clearly defined, but there also are several species complexes [41,42,43]. *Colletotrichum* spp. are causal agents of anthracnose in more than 3000 plant species and are among the top 10 fungal pathogens [44,45,46]. *C. truncatum*, *C. destructivum*, *C. coccodes*, *C. chlorophyte*, *C. gloeosporioides*, *C. incanum*, *C. plurivorum*, *C. sojae*, *C. musicola,* and *C. brevisporum* can all cause anthracnose on soybean [42,47,48,49,50,51,52,53,54]. Among those species *C. truncatum* is the most notorious [47] and there is relatively little information about the other species infecting soybean. Phylogenetic resolution is still in progress, with many studies addressing the issue [41,42,43,44,55,56,57,58,59,60].

Variation in the ITS region is not sufficient to discriminate all species, so Glyceraldehyde-3-Phosphate Dehydrogenase (*GAPDH*), *TUB2*, *CHS-1*, and *ACT* are used along with ITS, also when using with the soybean pathogens [42,55]. Among those *GAPDH* is most informative, at least for distinguishing the most important pathogen *C. truncatum* [61]. Based on ITS, three primers were designed and used to distinguish *C. gloeosporioides* and *C. truncatum* on soybean by performing classical multiplex-PCR [62]. The intergenic spacer (IGS) has been used as an alternative to ITS. An advantage is that it contains more polymorphic sites. It was efficiently used for detecting *C. lupini* in lupins by PCR and can be considered an alternative target for *Colletotrichum* species [63].

A multiplex TaqMan qPCR assay targeting the *GAPDH* gene was developed to detect and quantify *C. truncatum* along with *Corynespora cassiicola* and *S. sclerotiorum* in soybean seeds [64]. A multiplex qPCR assay targeting the *cox1* gene has also been established to distinguish the four soybean-infecting *Colletotrichum* species *C. chlorophyti*, *Glomerella glycines* (*Colletotrichum* sp.), *C. incanum*, and *C. truncatum*, by using two duplex sets. Successful detection was achieved with 0.1 pg of *C. truncatum* DNA, but when published, the assay had not yet been tested on host tissue samples [65].

For some *Colletotrichum* species that can infect soybean, diagnostic assays were developed because of damages on other host plants. These are *C. acutatum* on strawberries and grapevines [66,67], *C. coccodes* in soil and on potato tubers [68], *C. kahawae* on coffee [69], and *C. lagenarium* on cucurbit crops [70]. These assays may be transferred and used for diagnostics on soybean, too.

In addition to PCR and real-time PCR also LAMP (loop-mediated isothermal amplification) assays were established to detect *C. truncatum*, targeting the large subunit of RNA polymerase II (*Rpb1*) coding gene [71], and *C. gloeosporioides*, for which the target gene was a glutamine synthetase (*GS*) [72]. While these assays offer the advantage of diagnosis directly on the field because no PCR cycler is necessary and the reaction can be observed directly without any equipment, they are an order of magnitude less sensitive than the corresponding real-time PCR assays. For additional information on the mentioned primers and assays, see Table 4.

### 2.4. Genus Fusarium

Multiple *Fusarium* species are among the most important phytopathogenic and mycotoxigenic fungi [73,74]. Several *Fusarium* species are associated with soybean, causing *Fusarium* blight/wilt (*F. oxysporum*), sudden death syndrome (SDS, *F. virgiliforme* formerly *F. solani* f. sp. *glycines*), and root rot and seedling diseases (several *Fusarium* spp.) [1,75,76,77,78,79,80]. Diagnosis of the root pathogens may be difficult because they may either be the primary pathogen or infect together with other soilborne fungi (e.g., *Macrophomina*, *Phytophthora*, *Pythium*, and *Rhizoctonia*) [76,81]. *Fusarium* species have some differences in their housekeeping genes, and molecular identification has been widely used. Relevant genes for this genus are *TEF1*, *TUB*, mitochondrial small subunit rDNA (*mtSSU*), 28S rDNA, ITS, and IGS [82,83,84,85].

The causal agent of SDS, *F. solani* (Mart.) Sacc. f. sp. *glycines* [86], was first identified in 1989 [87,88]. At first, its identification has relied on morphological characteristics, which did not fully resolve the species complex. Using nuclear ribosomal DNA sequences of species within the *F. solani* complex, SDS-causing isolates were identified and *F. solani* f. sp. *phaseoli* was defined [82]. However, non-SDS-causing isolates are still included within *F. solani* f. sp. *phaseoli*. Another study using RAPD, could show that SDS-causing isolates form a cluster representing a biological subgroup within *F. solani* f. sp. *phaseoli*, and the authors suggested that it represents a separate *forma specialis* [89]. The differences between *F. solani* f. sp. *glycines* causing SDS and other *F. solani* are important for specific identification and detection. Differences in the *mtSSU* rRNA gene can be used to distinguish isolates of *F. solani* [83]. Detection of *F. solani* f. sp. *glycines* from plant and soil samples was enabled by a PCR assay using primers based on *mtSSU* and *TEF1* [83,90].

Two TaqMan probe qPCR assays for quantification of *F. virguliforme* from soybean plant samples based on *mtSSU* sequences are available [91,92]. Due to similarities in the *F. solani* species complex, the mitochondrial DNA (mtDNA) region is too conserved to differentiate *F. virguliforme* from the dry bean root rot pathogens *F. cuneirostrum* and *F. phaseoli* and other SDS causal agents, such as *F. tucumaniae*, *F. crassistipitatum*, and *F. brasiliense*, which dominate in South America [93,94]. However, the IGS region of the rDNA can resolve *F. virguliforme* from the other closely related *Fusarium* species, as demonstrated in multilocus genotyping studies of clade 2 *F. solani* species [93,94]. Following this, a TaqMan primer/probe set based on IGS to detect and quantify *F. virguliforme* in field-grown soybean roots and soil was established [95]. The *FvTox1* gene was also used to distinguish *F. virguliforme* from the other species within the SDS-BRR (bean root rot) clade in soil and soybean root samples with a species-specific TaqMan real-time qPCR assay [96].

Since the assay based on the *FvTox1* gene has a much higher limit of detection than assays based on the rDNA, yet another assay specific for *F. virguliforme* was developed, based on the IGS region [97]. This primer/probe set was later also used in a duplex qPCR assay for simultaneous detection of *F. virguliforme* and *F. brasiliense* [98]. Most recent (to our knowledge, 2022) is a series of primer/probe sets to detect *F. acuminatum*, *F. graminearum*, *F. proliferatum*, and *F. solani*, based on *TEF1* and *F. oxysporum*, and *F. equiseti,* based on IGS [99]. In this case, there are limits in specificity, especially of the *F. solani* primers/probe set (Fsol), which also amplifies *F. graminearum*, *F. equiseti*, and *F. virguliforme,* though much later than *F. solani*. For additional information on the mentioned primers and assays, see Table 5.

### 2.5. Genus Cercospora

Two species of genus *Cercospora* can infect soybeans. *C. kikuchii* (T. Matsumoto and Tomoy; M. W. Gardner) causes Cercospora leaf blight (CLB) and purple seed stain (PSS), while Frogeye leaf spot is caused by *C. sojina* Hara [1,100,101]. Production of a red toxin called cercosporin by *C. kikuchii* is recognized as a pathogenicity factor during colonization of soybean seeds and other aerial parts of the plant, including leaves, petioles, stems, and pods [102,103,104]. Cercosporin is also responsible for the symptoms of PSS: presence of purple spots against the natural color of the soybean seed coat [102] and causes membrane damage and cell death [105]. Cercosporin production is regulated by CFP (Cercosporin Facilitator Protein), which is specific for the genus and encoded by the gene *cfp* [106].

Seven nuclear gene regions and the mitochondrial (*cyb*) gene region were evaluated to study *Cercospora* species phylogenetically [107]. The seven regions included *ACT*, *CAL*, *HIS*, ITS, and *TEF1*, which were used in previous studies [108,109,110,111]. In addition, two primer pairs were designed based on *cfp* of *C. kikuchii* and one new primer Ck_Betatub-F1 to amplify *tub-1* after the complete *TUB* from *C. beticola* was analyzed [107]. *TUB* and *cfp* are excellent sources for polymorphic markers to investigate the relationships between the CLB and PSS pathogens.

The CNCTB6F/CNCTB6F primer pair, which targets the NADPH-dependent oxidoreductase gene (*CTB6*) from *C. nicotianae* [112], can be used to detect *C. kikuchii* and to differentiate between *C. kikuchii* and *C. sojina* [113]. The ITS1 and ITS2 sequences of genus *Cercospora* are too similar to develop primers for species-specific detection [113]. Therefore, a TaqMan real-time PCR assay was developed based on the *CTB6* gene to detect *C. kikuchii* [113] (Table 6).

### 2.6. Genus Septoria

*Septoria glycines* Hemmi causes Septoria brown spot, a foliar disease on soybean [1]. The pathogen infects pods and seeds but is rarely transmitted by seeds [114]. Early in the season, the symptoms are similar to those of bacterial blight (*Pseudomonas syringae* pv. *glycinea*) [115,116]. Later in the season, *Septoria* brown spot occurs together with frogeye leaf spot (*Cercospora sojina*) and *Cercospora* leaf blight (*C. kikuchii*) [117], making molecular diagnosis especially useful.

Three primers/probe sets were developed for qPCR based on *ACT*, *TUB*, and *CAL* [118]. The *CAL* set was not as specific as anticipated. The *ACT* set (Table 7) was specific to *S. glycines* for both conventional PCR and qPCR and the *TUB* set was specific only in qPCR.

### 2.7. Genus Macrophomina

The species in genus *Macrophomina* that is relevant to soybean is *Macrophomina phaseolina* (Tassi) Goid. This is one of the most severe soil and seed borne pathogens, attacking a wide range of hosts [119]. The fungus causes damping off, seedling blight, collar rot, stem rot, charcoal rot, and root rot diseases in various crops [120]. In soybean, *M. phaseolina* causes charcoal rot.

There is sequence variation between isolates of *M. phaseolina*. There have been several attempts to correlate this variation in several genetic markers with sampling region and host plant association, but while some groups found correlations [121], other publications could not [122,123,124], so that no *forma speciales* for soybean or another host plant have been defined, yet.

Consequently, the molecular detection assays that were established so far aim to detect all strains of *M. phaseolina*. In one approach that targets the ITS, the authors made sure to find primers on regions in the ITS that are conserved among *M. phaseolina* isolates, but different from other species [125]. The other approach to find a sequence conserved among *M. phaseolina* isolates was the sequence characterized amplified regions (SCAR) method. Using the universal rice primer URP-9F, a PCR product could be obtained that was the same for all *M. phaseolina* isolates and the resulting sequence (gene of unknown function) was used to design primers and probe for qPCR, enabling either SYBR green based or probe based qPCR [126]. A LAMP assay for detection of the species also uses the ITS sequence [127]. For additional information on the mentioned primers and assays, see Table 8.

### 2.8. Genus Phialophora

Brown stem rot (BSR) is a vascular disease caused by the soil-borne fungus *Phialophora gregata* f. sp. *sojae* (Allington and Chamberlain) Gams. This pathogen has two genotypes, “A” and “B” [128]. Genotypic differences among isolates correspond to phenotypic differences in the type and severity of symptoms. Isolates of genotype “A” are more aggressive than isolates of genotype “B” [129,130]. Genotypes “A” and “B” differ by a 188-bp insertion/deletion (INDEL) in the IGS of the ribosomal DNA and they also display cultivar preference [131,132,133].

Primers based on ITS were developed to identify *P. gregata* in infected soybean stems [134]. This primer pair was also used in combination with primer pair Plect1/Plect2 specific for *Plectosporium tabacinum*, to differentiate these two pathogens, which are associated with BSR [135]. Once the IGS region was found useful to distinguish “A” and “B”, primers BSRIGS1 and BSRIGS2 were designed [131]. In 2007, a qPCR assay was developed to quantify *P. gregata* f. sp. *Sojae* in plant tissue and in soil [136]. This qPCR assay does not yet distinguish between genotypes “A” and “B”. In 2009, a qPCR to specifically detect genotype “A” was developed. In combination with a specific primer/probe set [136], genotype “B” can also be quantified by determining the difference between the total *P. gregata* f. sp. *sojae* DNA amount and that of genotype “A” [137]. For additional information on the mentioned primers and assays, see Table 9.

### 2.9. Genus Rhizoctonia

*Rhizoctonia solani* Kühn (Teleomorph: *Thanatephorus cucumeris* (Frank) Donk) is a soil-borne fungal pathogen. *R. solani* is a species complex that was subdivided into anastomosis groups (AGs) based on an assessment of hyphal fusions [138,139]. AGs are subdivided into subgroups based on cultural morphology and physiological characteristics [140]. There are “13 AGs with 14 subgroups” [140,141,142]. The AGs vary in morphology, pathogenicity, and susceptibility to fungicides [143]. AG 1 to 4 cause disease in several economically important crops [138]. The other groups have more restricted host ranges or are less important. AG 12 is a special case: it forms mycorrhiza [141]. Molecular phylogenies have confirmed the AGs and the nomenclature was kept, even though the AGs may represent different species. The AGs important for soybean are AG 1-IA and AG 1-IB [1].

PCR detection assays were established for most of the AGs. Here, we present the assays for AG 1-IA and AG 1-IB. An assay based on ITS to detect AG 1-IA, AG 1-IB, and other AGs was designed in 2002 [144]. Later, two other groups [145,146] reported additional assays for the two AGs, respectively. As part of a real-time PCR study to detect and discriminate 11 AGs of *R. solani* using ITS regions, AG 1-IA was also detected [147]. In 2015, a LAMP assay was developed to detect *R. solani* in infected soybean tissues in the field [127]. For additional information on the mentioned primers and assays, see Table 10.

### 2.10. Genus Phakopsora

Soybean rust (SBR) is considered the economically most important disease on soybean. SBR is caused by two closely related fungi, *Phakopsora pachyrhizi* and *P. meibomiae* [148,149]. *P. pachyrhizi*, also called Asian soybean rust (ASR) since it originates from East Asia, is more aggressive and causes considerably greater yield loss [149]. The two species may be confused and early symptoms can be confused with bacterial pustule [1]. Therefore, molecular diagnosis of the soybean rust has been established. The ITS region has more than 99% nucleotide sequence similarity among different isolates of either *P. pachyrhizi* or *P. meibomiae*, but only 80% between the two species. Using the differences within the ITS region, four sets of primers were designed for *P. pachyrhizi* (Ppa1/Ppa2, Ppa3/Ppa4, Ppm1/Ppa2, and Ppm1/Ppa4) and two sets for *P. meibomiae* (Pme1/Pme2 and Ppm1/Pme2). The primers were tested and Ppm1/Ppa2 used as specific for *P. pachyrhizi* and Ppm1/Pme2 for *P. meibomiae* [150]. A VIC-labeled probe and the primers Ppm1/Ppm2 were designed as specific for genus *Phakopsora*.

*P. pachyrhizi* urediospores are wind-dispersed and, apart from diagnosis on plants, detection of spores in the air can be useful to predict epidemics or for scouting efforts. For this, the assay described above and another assay [151] were tested for sensitivity and from these assays another nested assay was created with a newly designed TaqMan probe (ITS1PhpFAM1). The nested assay combines the reverse primer Ppa2 specific to *P. pachyrhizi* [150] and a more general rust fungal forward primer ITS1rustF4a in the first round. In the second round ITS1rustF10d and ITS1rustR3d are combined with the probe. The assay can detect single and so is sensitive enough to find spores deposited in rain [152]. For additional information on the mentioned primers and assays, see Table 11.

### 2.11. Genus Phytophthora

*Phytophthora sojae* (Kaufm. and Gerd.) causes seed decay, root rot, damping off that may occur before or after emergence, stem rot, and sometimes foliar blight [153]. Soybean is the only major host of *P. sojae* [154]. Another Phytophthora species, *P. sansomeana*, has been isolated since 1981 from soybean in the USA and China [155,156,157,158,159].

The first target for molecular diagnosis was the ITS region. One PCR assay developed for *P. sojae* utilized primers PS1/PS2 [160]. The primers were also used in a SYBR-green based qPCR assay with a 10 pg limit of detection. Another group using the primers found problems with discrimination against other *Phytophthora* species from soybean [161]. Consequently, they developed their own PSOJF1/PSOJR1 primers, also targeting the ITS region [162]. Other researchers [163] using these primers reported a limit of detection of 10 fg in absolute quantifications. Other targets described for *P. sojae* detection are a Ras-related protein (*Ypt1*) coding gene [164] and an *A3aPro* transposon-like element [165].

A hierarchical approach to *Phytophthora* genus- and species-specific qPCR assays based on mitochondrial genes [166] provides new targets. Here, two loci were used, one for detecting all *Phytophthora* spp., the tRNA locus (*trnM-trnP-trnM*), and another, *atp9*, and the spacer between *atp9* and *nad9* (*atp9-nad9*) for genus- and species-specific detection. This approach was utilized to design specific probes for many *Phytophthora* spp. [167]. The system was also adapted for the isothermal technique recombinase polymerase amplification (RPA) [168]. Building on these approaches, a diagnostic assay for *P. sojae* and *P. sansomeana* was developed [169]. Using a genus specific probe and probes specific to *P. sojae* and *P. sansomeana* this multiplex qPCR assay can simultaneously determine if a sample is infected by any *Phytophthora* spp. and if it contains either *P. sojae*, *P. sansomeana*, or both [169]. The assay is highly specific and sensitive. A plant mitochondrial internal control for quantification relative to soybean and to determine the presence of PCR inhibitors can also be included in the assay. Another artificial internal control can be added when testing soil samples. Primer sets for RPA also exist [169].

Other groups [170,171] used the *Ty3/Gypsy* retroelement as target. This transposable element is widely distributed in the *Phytophthora* genus and forms lineages that predate the separation of the species [172]. This sequence is a good target because it is present in all isolates and has multiple copies per genome. Primers PS12 and PS6R were developed for this sequence and produced a 282 bp amplicon in all *P. sojae* isolates, but not on other *Phytophthora* spp. and other fungal soybean pathogens, as well as soybean itself [170]. This was developed into a probe-based qPCR assay for *P. sojae* [171]. For additional information on the mentioned primers and assays, see Table 12.

### 2.12. Genus Pythium

Of the other Oomycete genus, *Pythium*, up to 14 species have been reported to infect soybeans [173]. Importantly, *Pythium irregulare*, *P. sylvaticum*, *P. ultimum*, and *P. torulosum* cause seed rot, seedling damping-off, and root rot [174]. Species-specific primers for detection of *Pythium* spp. by PCR were developed [175,176,177]. For detection of *P. ultimum,* a LAMP assay is also available [178]. For the primers and assays, see Table 13.

## 3. General Steps and Procedures in Establishing Assays for Molecular Diagnosis

For readers who did not find a suitable assay to detect the pathogen they are interested in in Section 2 or elsewhere in the literature, here we provide a short and general overview over what is necessary to establish PCR tests for diagnosis of a given pathogen. In many cases, it may be useful if not necessary to follow most of these steps to establish detection of a pathogen in a new lab, even if primers for detection of a given pathogen were found. What we provide here cannot be a protocol, but we hope to be able to give important pointers on where to start and what are critical steps. Figure 2 gives an illustrative overview of the steps.

### 3.1. Sequence Determination and Primer Design

Molecular phylogenies for fungi are often based on sequences of a relatively small selection of genes or loci. These are the ITS (internal transcribed spacer) region, sequences in the rDNA that are spliced and do not contribute to the ribosome, *TUB* (tubulin), parts of the β-tubulin gene mostly consisting of introns [179,180], *TEF1* (translation elongation factor 1-α), also mostly introns, or calmodulin (*CAL*), histone-3 (*HIS*), also introns, or actin (*ACT*). These genes share the feature that they consist of highly conserved regions and variable parts with conservation in the exons and variability in the introns. The conserved regions make it possible that these genes can be amplified by PCR using conserved primers even from unknown species, while, on the other hand, the sequences obtained are variable enough to allow discrimination between the species [180]. Since these genes are all that is available as sequence information for many species, these also are the targets for PCR-based detection methods. In this case, primers are designed on the variable parts of the genes to obtain specific amplification.

In the special case that a detection method needs to be established, these sequences first need to be obtained. We recommend to stay with these genes since not only the primers (Table 14) and the PCR protocols are well established, but the obtained sequences will be valuable for phylogenetic classification of the species and also for these genes conserved and unique sections can easily be determined while these may be unknown for other genes. While this is our method of choice, it still should be mentioned that several of the assays presented in Section 2 are based on SCAR primers, designed to amplify sequences that were obtained by sequencing a RAPD fragment that is unique to a given species.

A very popular tool for the design of primers that are specific to one particular template is Primer-BLAST from National Center for Biotechnology Information (NCBI) [184]. Primer-BLAST allows either direct de novo design of new primers using Primer3 [185,186], that is implemented in Primer-BLAST, on a given template or checking of existing primers for specificity. The de novo design works quite well, but when sequences of closely related species are available (that should not be targets or that should be distinguished), it can be a better alternative to “manually” search for suitable primer positions in an alignment of sequences of these related species. In our own research we have used this latter option. For specificity checking, mostly the default settings can be kept (search mode, primer specificity stringency, max target amplicon size), but the database should be changed to nr. Here the search can be restricted to a range of non-target organisms. Working with soybean pathogens, the first non-target organism is soybean (*Glycine max*) itself, with all other microorganism growing on soybean next in line. With fungal pathogens, it could be a good idea to use Fungi as a taxonomic group for the whole range of species and *Pythium* and *Phytophthora* for the Oomycetes that could also occur. In the special case of probe-based qPCR, manual design of the probe might be preferred, especially because of the higher melting temperature of the probe. For specificity checking, a trick is necessary, since Primer-BLAST only allows for two primers. To get around this, the designated probe needs to be combined with forward and backward primers as two additional primer pairs and the Primer-BLAST output for all combinations needs to be compared.

Obviously, Primer-BLAST or other in silico approaches can only provide a prediction for the specificity of any primer pair. This prediction is limited by the alignment models that cannot be perfect and by the sequence databases that simply do not contain the full genome sequences of all soybean pathogens. Therefore, any primer pair or primer–probe combination also needs to be tested experimentally (Section 3.2). On the other hand, the databases are constantly growing and so also the predictions of Primer-BLAST are getting better and more comprehensive. Therefore, it may also be useful to use the tool to check the specificity of primers that were found in the literature.

### 3.2. In Vitro Test of the Primers for Efficiency and Specificity

Any predictions for specificity may be accurate or not, so they have to be tested experimentally. To test the primers, it is common to prepare DNA from pure cultures of different species and check whether using these DNA samples as template leads to any amplification. The first sample to be tested is the DNA prepared from the pathogen to be diagnosed. This template should be amplified. This test is repeated with different concentrations of the template. For classical PCR these experiments only yield a yes or no answer and a limit of detection based on DNA amount, but for qPCR also primer efficiency can be determined. Based on these results for a given primer pair or primer–probe combination, it needs to be decided whether the amplification is acceptable or not, or, if more than one primer pair was designed, the best one can be chosen.

Not to be forgotten is the no template control (NTC), which later on may indicate if there are contaminations in any of the reagents but importantly at the beginning of the testing indicates if the primers form dimers, hairpins, or other artifacts that were not predicted by the software used for primer design. If there is any product in the NTC that is not a contamination at this stage, other primers have to be obtained.

Then, a number of further DNA samples are tested, for which no amplification is expected. While, in contrast to the in silico predictions, these experimental tests give definite answers about the specificity of primers or primer-probe combinations, they may be quite laborious. Not only are there DNA preparations from several different samples, but first there have to be the different cultures, which may or may not be available in a lab. Often, isolates will have to be obtained from culture collections or from other labs. Therefore, these tests can be limited to soybean and other soybean pathogens. To further narrow down the selection, it can be restricted on the one hand to species that are closely related to the pathogen to be diagnosed, because with these species the risk for amplification by the chosen primers is highest, or on the other hand to pathogens that are also relevant on the tested samples. For example, if only seeds are tested, pathogens that are not found on or in seeds may be omitted or pathogens that only occur in different parts of the world can be regarded as irrelevant. While these restrictions are necessary because otherwise the testing may last for a very long time, in publications it should always be clearly communicated with which species the primers were tested, so that other researchers who may want to use the assay on a different tissue or in a different country know with which species they still may have to test the primers.

The latter also implies that most often primers found in literature have to be tested for specificity again. When primer pairs or primer–probe combinations are used in multiplex reactions, it is also necessary to check if the specificities that were determined for the separate PCR reactions are still valid in multiplex, since, theoretically, each primer in the multiplex can combine with any other primer in the mix. This necessity causes less work than it may seem, since the DNA samples needed for testing the multiplex are probably already there from testing the separate primer pairs or primer–probe combinations.

### 3.3. Test on Different Sample Types, Use of True Samples, Optimizing, Multiplexing

#### 3.3.1. Test on Different Sample Types

When primers are tested for efficiency, these experiments also yield a basis for quantification and at the same time also a limit of detection can be determined. However, because these experiments are performed with different amounts of DNA, the efficiency and the limit of detection also correspond to amount of DNA. Even though there are authors who are doing this, these values cannot easily be related to amounts of biomass. This is because even though the method or the scale can be adjusted to the amount of tissue being used for DNA preparation, the efficiency of DNA preparation is strongly dependent on the amount of biomass used for the preparation (learned from own experience, kits for DNA preparation give numbers for ideal sample weight). For example, if DNA prepared from a pure culture is diluted by 1:1000, the amount of DNA in a given volume is 0.1% of that in the same amount of undiluted DNA. However, if DNA is prepared from 0.1 mg instead of from 100 mg tissue, most likely no DNA at all is received and if DNA is received it is most probably not 0.1% of what would be obtained from 100 mg tissue. This means that especially limits of detection in ng DNA cannot be calculated into number of spores.

To more closely approximate what is seen in an actual plant sample, the fungal DNA can be diluted with soybean DNA instead of pure water. This may have a stabilizing effect on low concentrations of DNA and on the other hand introduce impurities with the soybean DNA that might inhibit the PCR reaction. To best simulate the actual assay, the soybean DNA should be from the same tissue that is also sampled, for example seeds, pods, stems, leaves, or roots. This allows to determine the extent of a pathogen infestation in ng pathogen DNA per ng soybean DNA prepared from a given sample [22].

However, to be able to determine the biomass of the pathogen in a given sample, standards have to be created where different amounts of pathogen are added to the intended sample material. This should be followed by DNA preparation and PCR or qPCR using this DNA.

This procedure can be nicely used with soil samples where different amounts of spores can be mixed with the soil. With a spore suspension and soil a relative uniform mix can be achieved, so that DNA prepared from a given amount of soil can be related with a number of spores and also the PCR or qPCR result (i.e., the Cq value) correlates to this number of spores. A watery suspension might be treated similarly, but, whereas soil contains DNA, this is not so for water, and this brings back the problems with DNA preparation efficiency described above. If the spores in question are not too tough, in this case adding the suspension directly into the PCR reaction without DNA preparation could be a better alternative.

Mixing defined amounts of fungal biomass with different soybean tissues is a difficult task. On the one hand, large amounts of both fungal biomass and plant tissue would be needed to achieve a mix with acceptable homogeneity and even if this could be achieved, the mix still might not adequately represent infected tissue in DNA preparation. Given these problems, it is best to accept that for plant samples PCR results cannot be related to a fungal biomass, but instead use ng fungal DNA per ng soybean DNA. Different groups have used different soybean genes for this kind of quantification and also as internal control. Examples for these genes and primers can be found in Table 15.

#### 3.3.2. Test of True Samples/Field Samples

Once the assay is established and also the foundations for quantification laid and a rough limit of detection is defined, it should be applied to actual sample material. This could be seed samples or plant material or soil samples collected from the field. It should be possible to detect the pathogen in these samples (if it is present), and the results obtained should be corroborated with classical methods. In some instances, this corroboration will not be entirely possible, since the molecular assay may allow for direct species identification or for quantification that are simply not possible with classical assays. In this case, it may be sufficient that the classical assay confirms the presence of a pathogen, while the qPCR defines the species and also quantifies it.

#### 3.3.3. Optimization

Diagnosis should be fast, sensitive, accurate, high throughput, and cheap. Speed is quite good for PCR methods, with 2–3 h to the result with classical PCR and 1–2 h with qPCR only counting the actual PCR. DNA preparation needs extra time.

Sensitivity and accuracy are mostly defined by the primers and/or probes. Therefore, optimization of these falls into Section 3.1 and Section 3.2. Here, different primer pairs or primer-probe combinations can be tested, and the ones with best selectivity and the highest amplification efficiency can be chosen. Both factors can also be influenced by changes in the PCR program, especially in chosen annealing temperature. Except for the actual PCR, sensitivity is also influenced by DNA preparation. Since the template amount limits detection, high efficiency in DNA preparation also leads to detection of lower amounts of pathogen. Additionally, polymerase inhibitors in the template DNA may be problematic. For soybean, in our own experience, pods and stems are rich in polymerase inhibitors. If these inhibitors lower the detection limit too much, it may be necessary to include additional steps into the DNA purification procedure to reduce the inhibitors.

Throughput generally is high with PCR methods and is further defined by sampling and DNA preparation. Sampling can make big differences if either many small samples are taken for each of which DNA is prepared or large amounts of tissue are pooled and homogenized and part of this material is used for DNA preparation. The latter option can dramatically reduce sample numbers and so increase throughput, but on the other hand homogenization of these large samples, for example many seeds together, can be difficult. Additionally, with larger samples, the results give a reduced resolution. In this issue throughput and the details of the results have to be balanced against each other.

The chemicals for PCR and especially qPCR can be quite expensive. However, there also are big differences between suppliers. Here it may be found that the cheaper product works quite well when compared to the more expensive one. Unfortunately, everything will have to be tested (again), since efficiencies and limits of detection are always defined for a given chemistry and need to be redefined for the alternative chemistry. Additionally, DNA preparation can be performed with different kits that may be expensive or cheap or just by classical protein precipitation followed by DNA precipitation. Often it can be found, that despite lower DNA yields or polymerase inhibitors, the sensitivity of the assay is still high enough despite using the cheapest method for DNA preparation.

Faster, cheaper, and easier, as it needs less specialized equipment than PCR, is LAMP. On the other hand, primer design for LAMP is much more difficult. So, instead of directly establishing LAMP for diagnosis of a pathogen, it may be a better option to establish a PCR diagnosis and then go for LAMP as an additional option, once the molecular diagnosis has shown its advantage. Right here, the authors have to state that they have had no personal experience with LAMP so far and, therefore, will not further discuss this technique.

#### 3.3.4. Internal Controls

Even when DNA preparation is optimized and the whole procedure is standardized, there may still be variation. PCR inhibitors may be present in some samples, but not in others, and can be especially problematic with soybean tissues or when testing soil. Additionally, any kind of handling mistakes or technical problems may occur at any stage of the qPCR process. This may lead to false negative results.

To meet this problem, it is possible to include internal controls. These are positive controls, reactions that should give an amplification in the PCR reaction if the reaction is working. When working with soybean tissue using a soybean gene as control target is a logical solution. Different groups have proposed different soybean genes as control targets (Table 15).

When searching for pathogens in soil or other environmental samples other than the host plant, no soybean DNA may be present, so the control targets mentioned above cannot be used. In these cases, it is possible to spike DNA preparations with target DNA, which may be added as genomic DNA from a pure culture of the pathogen, PCR product (either from the specific primers or using general primers), or as a plasmid. In the spiked reactions a positive outcome is expected; if this fails the presence of inhibitors or other technical problems is confirmed and other negative results may be considered false negatives. This spiking needs to be separately established for every pathogen. One internal control has been developed by a group working with soybean rust, which can theoretically be used for any pathogen [187]. The system consists of 111 nt random sequence with binding sites for primers and probe (Table 15). So, the DNA can be added as exogenous spiking material and the corresponding primers and probe to the qPCR reaction, either in a separate reaction or incorporated into the primary assay through multiplexing [187].

#### 3.3.5. Multiplexing

Showing so many PCR assays for diagnosis of different soybean pathogens in a review points to the possible advantages of combining these assays. Combining the assays by diagnosing different pathogens at the same time will reduce the number of necessary assays and, this way, reduce labor and costs. Indeed, combining PCR assays is possible by multiplexing. Multiplexing means that more than one PCR reaction is performed in the same PCR mix/the same reaction tube.

In classical PCR, this is achieved by simply combining different primer pairs with specificity to different pathogens in one reaction. Which pathogen was detected by the assay can be deduced from the bands on the gel on which the products were separated. To distinguish between different pathogens, it is necessary, however, that the different primer pairs in the reaction lead to amplicons of different sizes that can be recognized on the gel. This means that very often it is not possible to simply combine existing primer pairs in the same reaction, since they produce amplicons of similar size. So, for multiplexing additional efforts in primer design are necessary with strong limitations posed on the product sizes of the reactions. The number of different band sizes that can be distinguished on a gel also poses the limit of pathogens that can be detected at the same time.

In qPCR, multiplexing is realized by using probe mediated real-time PCR and using probes with different fluorophores for different pathogens. Amplicon size does not matter in this context, but, actually, the amplicons in a given multiplex real-time PCR reaction should have similar size since this influences PCR efficiency. In this case, the limit of multiplexing is defined by the number of different fluorophores that can be distinguished by the real-time PCR instrument. For many instruments, these are four or five.

As already mentioned in Section 3.2, any primer put into a multiplex reaction can theoretically pair up with any other primer in the mix to produce an amplicon. To avoid unwanted amplification extra care in primer design is necessary and, in case of qPCR, it must be ensured that all probes are different. Specificity needs to be tested in the multiplex (again). Combining assays from the literature may be possible, but is not very likely.

## 4. Potential and Challenges of Using Molecular Diagnosis and Establishing the Assays for Certification Purposes

Here, we want to summarize the advantages of PCR-based methods over classical methods, but also point to challenges that can be encountered.

### 4.1. Molecular Assays Are Fast and Yield Accurate Results

Most of this was already mentioned in the introduction. Probably the most important advantage of PCR based assays is identification of pathogens directly from infected tissues. For many fungi this is quite impossible with classical methods, since spores may or not be formed in the plant tissue and often also observation of colony morphology on the agar plate is necessary for species identification. Therefore, it may take weeks or even months to produce the structures by which a fungus can be identified. Sometimes, the process is further prolonged by the need to produce pure cultures. Additionally, sometimes, even then it is still impossible to reliably determine the species. Compared to this, PCR is very fast. Additionally, its specificity is not only high, but it can also be controlled.

### 4.2. qPCR Can Be Used to Quantify Pathogens, Enabling Methods to Test Strains for Aggressiveness or Cultivars for Resistance

Standard curves based on DNA dilutions from pure cultures of the pathogen or based on pathogen biomass mixed with soybean tissue or soil can be used for quantification using qPCR. As mentioned above (Section 3.3.1), it is important to know the limits of quantification.

The biomass of a pathogen in the soil can be a strong indicator for the danger that this pathogen poses for the crop. Since the methods are still relatively new, it is not yet clear what and how much can be learned from different amounts of pathogen in different plant tissues. For example, if a large amount of fungal DNA (seed borne pathogen, i.e., *Diaporthe* sp.) can be found in soybean seeds, are the plants more likely to die than if the seeds are only infected with little fungus? Unfortunately, it is not easy to make this connection, since the individual seeds for which the fungal biomass is determined are used in DNA preparation, so these seeds cannot be grown into plants to see how well this works.

Classical pathogenesis tests are based on inoculation of soybean with the pathogen. This can either be by seed inoculation, soil inoculation, or through wounds in the stem or on the nodes. Then, the symptoms are observed. To gain information about levels of resistance of different soybean cultivars, a long period of time may be required. Because of this, it can be a better alternative to inoculate plants or parts of plants, for example detached leaves, and closely thereafter quantify the pathogen in the inoculated tissue or in tissue adjacent to the inoculated tissue using qPCR. Based on how fast the pathogen grows, its aggressiveness or degrees of plant resistance can be determined. While these methods may be faster than the established procedures, it is also hard to establish them, however.

### 4.3. Tests for Certification Purposes Often Require Large Samples or Many Samplings

When a seed lot is tested for certification purposes, regulations require that a defined number or seeds are tested. This is necessary because the infestation of a seed sample is given as % infected seeds. If only few seeds are tested, the calculated percentage may be rather random. Only higher numbers of tested seeds that also should be randomly chosen can guarantee reliability of the resulting values. A common number that is used is 400 seeds per seed lot [188,189]. In this context, however, it needs to be stated that while the PCR methods are much faster than classical seed plating, they are more labor intensive. Preparing DNA from 400 soybean seeds individually and separately testing the DNA for presence of pathogen DNA is more work than placing the same 400 seeds on APDA plates and waiting for a fungus to grow out of them [188]. Unfortunately, to replicate the results obtained with the classical procedure, exactly this needs to be performed. If the 400 seeds are homogenized together and DNA is prepared from the resulting powder, no information is gained on how many of the seeds are infected, only a yes or no answer can be gained. The same is true if a method, such as seed soaking, is used. Here it may be possible to quantify the number of spores found in the soaking water, but that does not give the percentage of infected seeds either. On the other hand, the molecular method provides additional valuable information on the infecting species that may not be gained from visual inspection of the outgrowth of a seed plating test.

What should be established are procedures using the molecular method that still give information on the level of infestation. It is conceivable to correlate the percentage of infected seeds gained from doing DNA preparations for 400 seed separately with the amount of fungal DNA per plant DNA in a sample where seeds were homogenized together. This would reduce the amount of work, but still give information on the species that are present. If this should not be accurate enough, the large sample with 400 seeds could be combined with additional sampling of individual seeds, for example twelve. In any case, it must be realized that the results obtained with PCR cannot by 100% be matched with the results of the classical method. This also means that the regulations that were designed for the classical method should not be directly applied to the PCR methods. Instead, new regulations should be found that balance reduced sample sized against the additional information gained from molecular diagnosis. Combining classical seed plating with PCR methods would be yet another option and could yield the most comprehensive information.

## 5. Conclusions

As the reader will have found, there are lots of pathogens infecting soybean. Correspondingly, the number of molecular assays to detect them is large. We have striven to identify as many assays as possible and to present all that still have some relevance. If any assay is missing that should be there, this is an oversight and purely accidental. We hope that both our enumeration of available assays and our description of the establishment of an assay will prove useful.

## Figures and Tables

**Figure 1 jof-09-00587-f001:**
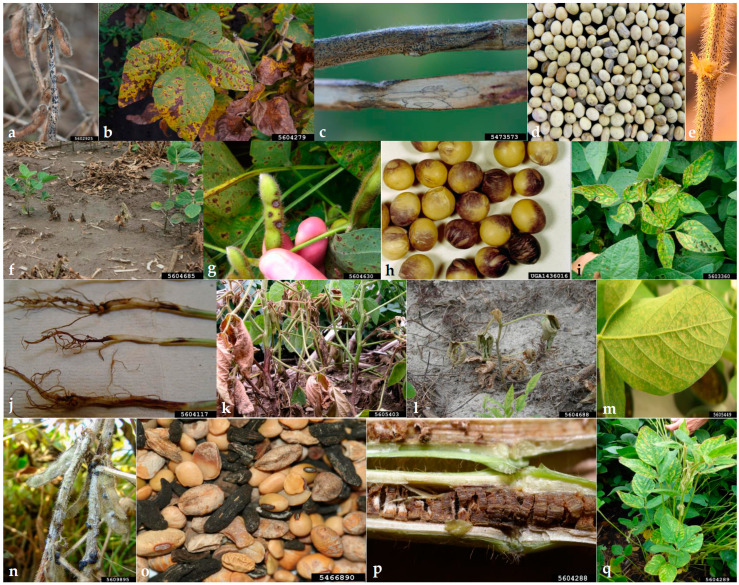
Soybean pathogens and diseases treated in this review. (**a**) Anthracnose (*Colletotrichum truncatum*), (**b**) Brown spot (*Septoria glycines*), (**c**) Charcoal rot (*Macrophomina phaseolina*), (**d**,**e**) Seed decay and pod and stem blight (*Diaporthe* spp.), (**f**) Damping-off and root rot (*Pythium aphanidermatum*), (**g**) Frogeye leaf spot (*Cercospora sojina*), (**h**) Purple seed stain (*Cercospora kikuchii*), (**i**) Sudden death syndrome (*Fusarium virguliforme*), (**j**) Fusarium root rot (*Fusarium* spp.), (**k**) Root and stem rot (*Phytophthora sojae*) (**l**) Rhizoctonia aerial blight (*Rhizoctonia solani*), (**m**) Asian soybean rust (*Phakopsora pachyrhizi*), (**n**) Sclerotinia stem rot (*Sclerotinia sclerotiorum*), (**o**) Sclerotia of *S. sclerotiorum* in soybean seed lots, (**p**,**q**) Brown stem rot (*Phialophora gregata*). Images (**a**,**o**) Daren Mueller, Iowa State University, Bugwood.org; (**b**) Craig Grau, Bugwood.org; (**c**) Martin Draper, USDA–NIFA, Bugwood.org, (**d**,**e**) Behnoush Hosseini, University of Hohenheim (original); (**f**) Martin Chilvers, Bugwood.org; (**g**) Trey Price, LSU AgCenter, Bugwood.org; (**h**) Clemson University—USDA Cooperative Extension Slide Series, Bugwood.org; (**i**) Kiersten Wise, Bugwood.org; (**j**) Loren Giesler, University of Nebraska, Bugwood.org; (**k**) Craig Grau, Bugwood.org; (**l**,**q**) Tristan Mueller, Bugwood.org; (**m**,**n**) Gerald Holmes, Strawberry Center, Cal Poly San Luis Obispo, Bugwood.org; (**p**) Alison Robertson, Bugwood.org.

**Figure 2 jof-09-00587-f002:**
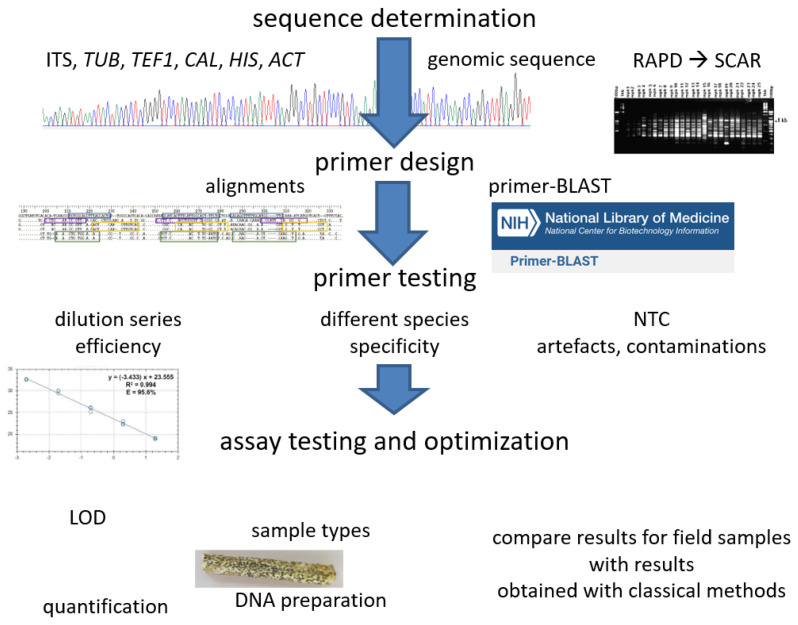
Illustrative overview on the procedures of establishing a qPCR method for pathogen detection. All details can be found in the main text. (Own design, generic illustrations from own pictures).

**Table 1 jof-09-00587-t001:** Important fungal and oomycete pathogens causing soybean diseases (selection from [1]).

Common Name	Causal Organisms
Anthracnose	*Colletotrichum truncatum*
Brown spot	*Septoria glycines*
Charcoal rot	*Macrophomina phaseolina*
Pod and stem blight	*Diaporthe sojae*, *Diaporthe* spp.
Phomopsis seed decay ^1^	*Diaporthe longicolla*, *Diaporthe sojae*, *Diaporthe* spp.
Stem canker	*Diaporthe caulivora* ^2^, *Diaporthe aspalathi* ^3^, *Diaporthe* spp.
Pythium damping off and root rot	*Pythium ultium*, *P. aphanidermatum*, *P. irregolare*, *P. torulosum*
Frogeye leaf spot	*Cercospora sojina*
Fusarium root rot	*Fusarium* spp.
Phytophthora root and stem rot	*Phytophthora sojae*
Purple seed stain/Cercospora leaf blight	*Cercospora kikuchii*
Sudden death syndrome	*Fusarium virguliforme*
Rhizoctonia aerial blight	*Rhizoctonia solani*
Asian soybean rust	*Phakopsora pachyrhizi*
Sclerotinia stem rot (white mold)	*Sclerotinia sclerotiorum*
Brown stem rot	*Phialophora gregata*

^1^ Petrovic et al. [7] proposed that Phomopsis seed decay be called Diaporthe seed decay (DSD). ^2^ Santos et al. [8] shortened the name of *Diaporthe phaseolorum* var. *caulivora* to *D. caulivora* and, at the same time, also proposed that it should be considered as a separate species. ^3^ The previous name of *Diaporthe aspalathi* was *D. phaseolorum* var. *meridionalis* [8].

**Table 2 jof-09-00587-t002:** Primers and corresponding assays for the diagnosis of *Diaporthe* spp. on soybean.

Target Gene	Target Species (Specificity)	Primer/Probe (Combination)	Sequence (5′-3′)	T_m_ (°C)	Assay	Ref.
ITS	*Diaporthe* sp., but based on *D. phaseolorum* and *D. longicolla*	Phom.I	GAGCTCGCCACTAGATTTCAGGG	60	PCR	[15]
		Phom.II	GGCGGCCAACCAAACTCTTGT			
ITS	*D. aspalathi*	DphLe	TCGGCCTTGGAAGTAGAAGAC	60	PCR	[20]
		DphRi	ACTGAATGCGTTGCGATTCT			
ITS	*D. caulivora*	DPC-3F	TTTATGTTTATTTCTCAGAGTTTCAGTGTAA	60	qPCR	[21]
		DPC-3R	GGCGCACCCAGAAACC			
		DPC-3P	FAM-CGGGCTGCTCCCCGTCTCC-TAMRA			
ITS	*D. longicolla*	PL-3F	CAGAGATTCACTGTAGAAACAAGAGTTT	60	qPCR	[21]
		PL-3R	CCGGCCTTTTGTGACAAA			
		PL-3P	FAM-CGGGCTGCTCCCTGTCTCCAG-TAMRA			
ITS	*D. longicolla*, *D. aspalathi*, *D. sojae*	PL-5F	CGAGCTCGCCACTAGATTTCA	60	qPCR	[21]
		PL-5R	CCTCAAGCCTGGCTTGGTGATGG			
		PL-5P	FAM-CCATCACCAAGCCAGGCTTGAGG-TAMRA			
*TEF*	*D. longicolla*	DPCL-F	TGTCGCACCTTTACCACTG	60	qPCR ^a^	[22]
		DPCL-R	GAACGATCCAAAAAGCTCTC			
		DPCL-P	FAM-GCATCACTTTCATTCCCACTTTCTG-BMN-Q535			
*TEF*	*D. caulivora*	DPCC-F	GCCTGCAAAACCCTGTTAC	60	qPCR ^a^	[22]
		DPCC-R	CATCATGCTTTAAAAATGGGG			
		DPCC-P	Cy5-CTCTTACCACACCTGCCGTCG-BMN-Q620			
*TEF*	*D. eres*	DPCE-F	ACTCACTCAATCCTTGTCAC	60	qPCR ^a^	[22]
		DPCE-R	GAGGGTCAGCATAATATTCG			
		DPCE-P	ROX-CCATCAACCCCATCGCCTCTTTC-BMN-Q590			
*TEF*	*D. novem*	DPCN-F	AAAACCCTGCTGGCATTAAC	60	qPCR ^a^	[22]
		DPCN-R	TATTCTTGACAGTTCGTTTCG			
		DPCN-P	HEX-TCTACCACTTTCAACCCTATCAATC-BMN-BMN-Q535			

^a^ Probe based real-time PCR. These four primer-probe-combination were designed for use together in a quadruplex reaction.

**Table 3 jof-09-00587-t003:** Primers and corresponding assays for diagnosis of *Sclerotinia sclerotiorum* on soybean.

Target Gene	Primer/Probe (Combination)	Sequence (5′-3′)	T_m_ (°C)	Fragment Length (bp)	Assay	Ref.
ITS	SSFWD	GCTGCTCTTCGGGGCCTTGTATGC	65 ^a^	278	PCR and qPCR	[35]
	SSREV	TGACATGGACTCAATACCAAGCTG				
ITS	M13FWD	GTAAAACGACGGCCAGT		252	qPCR ^b^	[37]
	M13REV	CAGGAAACAGCTATGAC				
*mitochodrial small rRNA*	mtSSFor	AGGTAACAAGTCAGAAGATGATCGAAAGAGTT		125/80?	qPCR	[39]
	mtSSRev	GCATTAAGCCTGTCCCTAAAAACAAGG				
SS1G_00263	SSBZF	GCTCCAGCAGCCATGGAA	60		qPCR	[40]
	SSBZR	TGTTGAAGCAGTTGACGAGGTAGT				
	SSBZP	CAGCGCCTCAAGC				

^a^ Used in touchdown PCR. ^b^ Eva Green based real-time PCR. These four primer pairs can be used together in two duplex reactions where the products are distinguished by their melting temperature.

**Table 4 jof-09-00587-t004:** Primers and corresponding assays for diagnosis of *Colletotrichum* spp. On soybean.

Target Gene	Target Species (Specificity)	Primer/Probe (Combination)	Sequence (5′-3′)	T_m_ (°C)	Fragment Length (bp)	Assay	Ref.
*cox1*	*C. chlorophyti*	cox1AF	CCTGGTATAAGATTACATAAG	55	115	qPCR ^a^	[65]
		cox1AR	CTGTAAGTACCATAGTAATTG				
*cox1*	*C. sojae*	cox18AF	ACATTTATCAGGAGTAAGTAG	55	77	qPCR ^a^	[65]
		cox18AR	TTCCAGGTGTTCTCATAT				
*cox1*	*C. incanum*	cox6AF-2	ATGAACATTATATCCTCCTT	55	115	qPCR ^a^	[65]
		cox6AR-2	ATTAACTGCTCCTAATAAAC				
*cox1*	*C. truncatum*	cox15BF	TTATGCCAGCCTTAATAG	55	117	qPCR ^a^	[65]
		cox15BR	AAGATGGTGGTAATAATCA				
ITS	*C. gloeosporioides*	Colg 1	AACCCTTTGTGAACATACC	63	443	qPCR	[62]
		Colg 2	CCCTCCGGATCCCAG				
ITS	*C. truncatum*	Colg 1	AACCCTTTGTGAACATACC	63	375	qPCR ^a^	[62]
		CT 2	CTTTAAGGGCCTACGTCAA				
ITS	*C. acutatum*	CaITS_F701	GGATCATTACTGAGTTACCGC	60	80	qPCR	[66]
		CaITS_R699	GCCCGCGAGAGGCTTC				
		CaITS_R815 ^b^	GCCCACGAGAGGCTTC				
		CaITS_P710	TACCTAACCGTTGCTTCGGCGGG				
ITS	*C. acutatum*	ACUT-F1	CGGAGGAAACCAAACTCTATTTACA	60	70	qPCR	[67]
		ACUT-R1	CCAGAACCAAGAGATCCGTTG				
		ACUT-PB	CGTCTCTTCTGAGTGGCACAAGCA				
ITS	*C. gloeosporioides*	GLOE-F1	GGCGGGTAGGGTCYCCG	60	101	qPCR	[67]
		GLOE-R2	ACTCAGAAGAAACGTCGTTAAATCAG				
		GLOE-PB	CTCCCGGCCTCCCGCCYC				
ITS	*Colletotrichum* sp.	COL GEN-F1	TGCCTGTTCGAGCGTCATT	60	111	qPCR	[67]
		COL GEN-R2	CTACGCAAAGGAGGCTCCG				
		COL GEN-PB	AACCCTCAAGCWCYGCTTGGYKTTGG				
IGS	*C. lupini*	CLF	CCCGAGAAGGCTCCAAGTA	63		PCR	[63]
		CLR	CATAAACGCCTAAGAACCGC				
*GAPDH*	*C. truncatum*	ColT-F6	TTGAGACCAAGTACGCTGTATGTATCAC	60		qPCR	[64]
		ColT-R5	TTCTGCCTCACATCGAACTCTC				
		ColT-P	HEX-CAGCCTTCG/ZEN/ACTCTCGTTGGAAAA-IABkFQ				
*GS*	*C. gloeosporioides*	F3	GCTGCAGCCGGAAAATCC	64		LAMP	[72]
		B3	GGCAGACTCGGAGAGACC				
		FIP (F1c + F2)	ACCGGCTCAGCTGCAACGC-ACACGAGCAAAAGGATACGC				
		BIP (B1c + B2)	TAATGCCTTTCACGACCTGCGG-CCGAGGCAATGATTCCTCAA				
		LF	CGGGCCAACGCTGGAAAA				
		LB	GGCGCAACAAAGCTGGG				
*Rpb1*	*C. truncatum*	F3	ACGGAGAATACTCTCTGGGT	62		LAMP	[71]
		B3	AGGATGTTGTGTGCCATCTC				
		FIP (F1c + F2)	GCCTTGTGTCGGACTCTGGG-GCAAGCTCCCGTTAACCA				
		BIP (B1c + B2)	ACAGCTTGTCGCCAAGTACGAG-GGGTGTGATCTGAGGCTCTT				
		LF	TGAATGTTGCCACAGCCGC				

^a^ Eva Green based real-time PCR. These four primer pairs can be used together in two duplex reactions where the products are distinguished by their melting temperature. ^b^ Alternative reverse primer to cover intraspecific sequence variation.

**Table 5 jof-09-00587-t005:** Primers and corresponding assays for diagnosis of *Fusarium* spp. on soybean.

Target Gene	Target Species (Specificity)	Primer/Probe (Combination)	Sequence (5′-3′)	T_m_ (°C)	Fragment Length (bp)	Assay	Ref.
ITS1	*F. solani* f. sp. *phaseoli*	FspF	ACCCCCTAACTCTTGTTATATCC	60	957–958	PCR	[82]
		FspR	GCGCAATACCCTGAGGCG				
*TEF1*	*F. solani* f. sp. *phaseoli*	Effp-1	AACCCCGCCCGAGGACTCA	72	562	PCR/qPCR	[84]
		Effp-2	AGACATGAGCGATGAGAGGCA				
*TEF1*	*F. solani* f. sp. *glycines*	FsgEF1	GAGTCGGTTAGCTTCTGTC	66 ^a^/56	237	PCR ^b^	[90]
		FsgEF2	GCGCGCCTTGCTATTCTCC				
*mtSSU*	*F. solani* f. sp. *glycines*	Fsg1	GTCTTCTAGGATGGGCTGGT	66 ^b^/56	438	PCR ^c^	[90]
		Fsg2	CATTTAATGCCTAGTCCCCTATCA				
*mtSSU*	*F. solani* f. sp. *glycines*	Fsg-q-1F	GATACCCAAGTAGTCTTTGCAGTAAATG	60		qPCR	[91]
		Fsg-q-1R	TTAATGCCTAGTCCCCTATCAACAT				
		Fsg-q-1P	6FAM-TGAATGCCATAGGTCAGAT-MGBNFQ				
*mtSSU*	*F. solani* f. sp. *glycines*	FSGq1	AACCCTTTGTGAACATACC	60		qPCR	[92]
		FSGq2	CCCTCCGGATCCCAG				
		FSG-MGB probe	6FAM-TCTTCTAGGATGGGCTGGT-MGBNFQ				
*FvTox1*	*F. virguliforme*	FV-F	GCAGGCCATGTTGGTTCTGTA	60	200	qPCR	[96]
		FV-R	GCACGTAAAGTGAGTCGTCTCATC				
		FV-MGB probe	6FAM-ACTCAGCGCCCAGGA-MGBNFQ				
IGS	*F. virguliforme*	FvIGS-F1	GGTGGTGCGGAAGGTCT	66		qPCR	[95]
		FvIGS-R3	CCCTACACCTTTCGTACCAT				
		FvIGS-Probe2	6FAM-ATAGGGTAGGCGGATCTGACTTGGCG-TAMRA				
IGS	*F. virguliforme*	F6-3	GTAAGTGAGATTTAGTCTAGGGTAGGTGAC	60		qPCR	[97]
		R6	GGGACCACCTACCCTACACCTACT				
		FvPrb-3	6FAM-TTTGGTCTAGGGTAGGCCG-MGBNFQ				
IGS	*F. brasiliense*	Fb_F2	AGGTCAGATTTGGTATAGGGTAGGTGAGA	67 ^f^	130	qPCR ^d^	[98]
		Fb_R2	CGGACCATCCGTCTGGGAATTT	66 ^f^			
		Fb_Prb1	5HEX-TGGGATGCCCT+AATTTTT+ACGG-3IABkFQ ^e^	65 ^f^			
*TEF1*	*F. acuminatum*	FacuF	TCGCGCACTACATGTCTT	54 ^g^	142	qPCR	[99]
		FacuR	AGAGAGCGATATCAATGGTGA	53 ^g^			
		FacuP	FAM-AACCACTGG/ZEN/ACAATAGGAAGCCGC	61 ^g^			
*TEF1*	*F. graminearum*	FgraF	CTCTTCCCACAAACCATTCC	53 ^g^	104	qPCR	[99]
		FgraR	TACTTGAAGGAACCCTTACC	51 ^g^			
		FgraP	FAM-ACCACCTGT/ZEN/CAATAGGAAGCCGCC	63 ^g^			
*TEF1*	*F. proliferatum*	FproF	GCGTTTTTGCCCTTTCCTGT	57 ^g^	123	qPCR	[99]
		FproR	AACCCAGGCGTACTTGAAGG	57 ^g^			
		FproP	FAM-AGGAAGCCG/ZEN/CTGAGCTCGGT	64 ^g^			
*TEF1*	*F. solani*	FsolF	AAACCCTCATCGCGATCTG	55 ^g^	108	qPCE	[99]
		FsolR	AGTGACCGGTCTGTAGATGA	55 ^g^			
		FsolP	FAM-CCTGGTATC/ZEN/TCGGGCGGG	60 ^g^			
IGS	*F. equiseti*	FequiF	TGTTGGGACTCGCGGTAA	56 ^g^	94	qPCR	[99]
		FequiR	GATTACCAGTAACGAGGTGTA	51 ^g^			
		FequiP	FAM-CACGTCGAG/ZEN/CTTCCATAGCGTAGT	60 ^g^			
IGS	*F. oxysporum*	FoxyF	CCGTCGATAGGAGTTCCGTC	57	80	qPCR	[99]
		FoxyR	TCGAACCGACCATCTCCAAG	57			
		FoxyP	FAM-TGGACGGTG/ZEN/CAGGGTAGG	64			

^a^ First is the first melting temperature in the touchdown program, the other is the last. ^b^ For optimal specificity, a protocol for touchdown PCR is used. For optimal sensitivity, a nested PCR is described using additional general *TEF1* primers for the first round. ^c^ For optimal specificity, a protocol for touchdown PCR is used. For optimal sensitivity, a nested PCR is described using additional general mtDNA primers (NMS) for the first round. ^d^ The assay was designed for duplexing with the primer/probe set for *F. virguliforme* above. ^e^ The + sign indicates that the next base is a locked nucleic acid (LNA) residue. These are necessary because the probe covers only two nucleotides that differ between *F. brasiliense* and other *Fusarium* spp., i.e., *F. virguliforme*. Do not be discouraged when checking Figure 1 in [98]; there the probe sequence is given wrongly and the position is off by one base, but the sequence given here is correct. ^f^ Calculated primer melting temperatures. The actual annealing temperature used was not reported. ^g^ Calculated primer melting temperatures. All these assays were run with an annealing temperature of 60 °C.

**Table 6 jof-09-00587-t006:** Primers and corresponding assays for diagnosis of *Cercospora kikuchii* on soybean.

Target Gene	Primer/Probe (Combination)	Sequence (5′-3′)	T_m_ (°C)	Fragment Length (bp)	Assay	Reference
*CTB6*	CKCTB6-2F	CACCATGCTAGATGTGACGACA			qPCR	[113]
	CKCTB6-2R	GGTCCTGGAGGCAGCCA				
	CKCTB6-PRB	CTCGTCGCACAGTCCCGCTTCG				

**Table 7 jof-09-00587-t007:** Primers and corresponding assay for diagnosis of *S. glycines* on soybean.

Target Gene	Primer/Probe (Combination)	Sequence (5′-3′)	T_m_ (°C)	Fragment Length (bp)	Assay	Reference
*ACT*	Ac1(f)	ACAATCCAGGGACCACAATC	60 ^a^	90–100	qPCR	[118]
	Ac2(r)	ATGGCTGATCGCATACCC	59 ^a^			
	Ac(probe)	6FAM-AGAGCTGACCAGGACCCAGCATCCA-TAM	73 ^a^			

^a^ Calculated primer melting temperatures. The assay was run with an annealing temperature of 60 °C.

**Table 8 jof-09-00587-t008:** Primers and corresponding assays for diagnosis of *M. phaseolia*.

Target Gene	Primer/Probe (Combination)	Sequence (5′-3′)	T_m_ (°C)	Fragment Length (bp)	Assay	Ref.
ITS	MpKFI	CCGCCAGAGGACTATCAAAC	56	350	PCR	[125]
	MpKRI	CGTCCGAAGCGAGGTGTATT				
	MpKH1 ^a^	GCTCTGCTTGGTATTGGGC	55 ^a^		dot blot	
*Ukn* ^b^	MpSyK F	ATCCTGTCGGACTGTTCCAG	60		qPCR	[126]
	MpSyK R	CTGTCGGAGAAACCGAAGAC				
	MpTqK F	GCCTTACAAGGGTCTCGTCAT	60		qPCR	
	MpTqK R	CCCTTGGCGATGCCGATA				
	MpTqK P	6-FAM-CAGGCCACAGGATCTT-MGBNFQ				
ITS	F3	GCACATTGCGCCCCTTG	62 ^c^		LAMP	[127]
	B3	GTTCAGAAGGTTCGTCCGG				
	FIP	AGGACGGTGCCCAATACCAAGCGGGGCATGCCTGTTCGA				
	BIP	CTCAAAGACCTCGGCGGTGGGCTCCGAAGCGAGGTGTA				

^a^ DIG labeled probe. Correspondingly also the temperature is the hybridization temperature. ^b^ Sequence obtained using the SCAR approach. ^c^ Incubation for the LAMP assay. Incubation for 60 min.

**Table 9 jof-09-00587-t009:** Primers and corresponding assays for diagnosis of *P. gregata* f. sp. *sojae*.

Target Gene	Primer/Probe (Combination)	Sequence (5′-3′)	T_m_ (°C)	Fragment Length (bp)	Assay	Ref.
ITS	BSR1	GCTTGCTCCGTGGCGGGCTG	60	480	PCR	[134]
	BSR2	AATTTGGGTGTTGCTGGCATG				
IGS	BSRIGS1	GGGGTTCCGGGATTCACAGG	55	1020 ^a^	PCR	[131]
	BSRIGS2	GAGTGGTAAATGGGGTAATCAAC		830 ^a^		
IGS	BSRqPCRf1	CAAACCAGGGCCGATCAG	60		qPCR	[136]
	BSRqPCRr1	CGGATTCAGCGTAAAAAATGG				
	BSRqPCRpb1	6-FAM-CTCCCGTATGGTTTCT-MGBNFQ				
IGS ^b^	PgsAspF	GGAATTGGTGGGAGAGG	60	69	qPCR	[137]
	PgsAspR	GACTTCTAGGGTATGTCTACAGTG				
	PgsAspPR	CAL Flour Red 610-AGGCTACTCTTACAGGCTCTC-BHQ-2				

^a^ The PCR product is 1020 bp for genotype A and 830 bp for genotype B. This is how this assay can distinguish the two genotypes. ^b^ Target in this special case is the INDEL sequence specific for genotype A. It is impossible to get primers specific for genotype B using the same strategy, which is why for quantification of genotype B these primers are used together with the assay by [136].

**Table 10 jof-09-00587-t010:** Primers and corresponding assays for diagnosis of *R. solani* on soybean.

Target Gene	Target Species (Specificity)	Primer/Probe (Combination)	Sequence (5′-3′)	T_m_ (°C)	Fragment Length (bp)	Assay	Ref.
28S ribosomal DNA	*R. solani*	AG-common f	CTCAAACAGGCATGCTC	54		PCR	[144]
	AG 1-IA	AG 1-IA	CAGCAATAGTTGGTGGA		265		
	AG 1-IB	AG 1-IB	AAGGTCCTTTGGGGTTGGGG		300		
*Unk* ^a^	AG 1-IB	N18-rev	AGCGTGCTAACATAGTCACTC		324	PCR	[145]
		N18-for	ACACTAGAGTAGGTGGTATCA				
ITS	AG 1-IA	Rs1F	GCCTTTTCTACCTTAATTTGGCAG	60	137–140	qPCR	[146]
		Rs2R	GTGTGTAAATTAAGTAGACAGCAAATG				
ITS1	AG 1-IA	AG-1-1A_F	TTGTTGCTGGCCTTTTCTACCT	60		qPCR	[147]
		AG-1-1A_R	ATGGAATTAAATCCACCAACTATTGCT				
		AG-1-1A_P	FAM-CATCACACCCCCTGTGCACTTGTGAGA-TAMRA				
ITS	*R. solani*	F3	CGAAATGCGATAAGTAATGTGAA	62 ^b^		LAMP	[127]
		B3	AGAGGAGCAGGTGTGAAG				
		FIP	GCTCCAAGGAATACCAAGGAGCCAGAATTCAGTGAATCATCGAATC				
		BIP	TGCCTGTTTGAGTATCATGAATTCTAAAAGACCTCCAATACCAAAG				

^a^ Sequence obtained by sequencing a RAPD fragment. SCAR method for primer identification. ^b^ Incubation for the LAMP assay. Incubation for 60 min.

**Table 11 jof-09-00587-t011:** Primers and corresponding assays for diagnosis of *P. pachyrhizi and P. meibomiae* on soybean.

Target Gene	Target Species (Specificity)	Primer/Probe (Combination)	Sequence (5′-3′)	T_m_ (°C)	Fragment Length (bp)	Assay	Ref.
ITS	*P. pachyrhizi*	Ppm1	GCAGAATTCAGTGAATCATCAAG	65 ^a^ 60 ^b^	141	PCR/qPCR	[150]
		Ppa2	GCAACACTCAAAATCCAACAAT				
ITS	*P. meibomiae*	Ppm1	GCAGAATTCAGTGAATCATCAAG		139		
		Pme2	GCACTCAAAATCCAACATGC				
ITS	*Phakopsora*	Ppm1	GCAGAATTCAGTGAATCATCAAG		77		
		Ppm2	CTCAAACAGGTGTACCTTTTGG				
ITS	*Phakopsora*	FAM-probe ^c^	FAM-CCAAAAGGTACACCTGTTTGAGTGTCA-TAMRA				
		VIC-probe ^d^	VIC-TGAACGCACCTTGCACCTTTTGGT-TAMRA				
ITS	*P. pachyrhizi*	ITS1rustF4a ^e^	GAGGAAGTAAAAGTCGTAACAAGGTTTC	60		nested qPCR	[151,152]
		ITS1rustF10d	TGAACCTGCAGAAGGATCATTA				
		ITS1rustR3d	TGTGAGAGCCTAGAGATCCATTG				
		ITS1PhpFAM1	FAM-TCATTGAT-TGATAAGATCTTTGGGCAATGG-3IABlkFQ				

^a^ With standard PCR. ^b^ With qPCR. ^c^ For use with Ppm1/Ppa2 and Ppm1/Pme2. ^d^ For use with Ppm1/Ppm2. ^e^ Combined with Ppa2 from [150] in first round of nested PCR.

**Table 12 jof-09-00587-t012:** Primers and corresponding assays for diagnosis of *Phytophthora* spp. on soybean.

Target Gene	Target Species (Specificity)	Primer/Probe (Combination)	Sequence (5′-3′)	T_m_ (°C)	Fragment Length (bp)	Assay	Ref.
ITS	*P. sojae* ^a^	PS1	CTGGATCATGAGCCCACT	66	330	PCR/qPCR ^b^	[160]
		PS2	GCAGCCCGAAGGCCAC				
ITS	*P. sojae*	PSOJF1	GCCTGCTCTGTGTGGCTGT	50	127	qPCR ^b^	[162]
		PSOJR1	GGTTTAAAAAGTGGGCTCATGATC				
*Ypt1*	*P. sojae*	F3	CCTTGTCTGCCCTCTCGA	65 ^b^		LAMP	[164]
		B3	AGAAGCGTACACCCACCA				
		FIP	GAATTTTCTGGGCGGGACAACGCCAGGATGGCTAAGGTTTCC				
		BIP	GAGCTGGACGGCAAGACCATCCATAAGTGCGCTTAACCGG				
		LF	GCACAATATTGTCAGCAACTGGATC				
		LB	CAAGCTCCAGATTGTACGTTCA				
*A3aPro*	*P. sojae*	F3	GCGTATTGAGGGTTGCTG	64 ^c^		LAMP	[165]
		B3	GCGTCCTATCACCTAGTGC				
		FIP	ACGTGGGTTCGGATTGGACC-CTTGGGTACTGTGTACCAG				
		BIP	CGCCACCGATGATTCGACGA-AATCAACCATCACTCACCG				
		LB	GTAGGATGATTGGATGAACAC				
*atp9*	*Phytophthora*	PhyG_ATP9_2FTail	AATAAATCATAACCTTCTTTACAACAAGAATTAATG	57		multiplex qPCR	[166,169]
*nad9*		PhyG-R6_Tail	AATAAATCATAAATACATAATTCATTTTTATA				
*atp9-nad9*		Phytophthora genus-specific TaqMan probe	FAM-AAAGCCATC [ZEN] ATTAAACARAATAAAGC-IABkFQ				
*atp9-nad9*	*P. sojae*	P. sojae species-specific TaqMan probe	HEX-TTGATATAT [ZEN] GAATACAAAGATAGATTTAAGTAAAT-IABkFQ				
*atp9-nad9*	*P. sansomeana*	P. sansomeana species-specific TaqMan probe	Quasar670-TATTAGTACTAAYTACTAATATGCATTATTTTTAG-BHQ-2				
*tRNA-M*	*Phytophthora*	TrnM-F	ATGTAGTTTAATGGTAGAGCGTGGGAATC	39 ^d^		RPA	[168,169]
		TrnM-R	GAACCTACATCTTCAGATTATGAGCCTGATAAG				
		TrnM-P	TAGAGCGTGGGAATCATAATCCTAATGTTG [FAM-dT] A [THF] G [BHQ1-dT] TCAAATCCTACCATCAT [3′-C3SPACER]				
*atp9*		Atp9-F	CCTTCTTTACAACAAGAATTAATGAGAACCGCTAT				
*atp9-nad9*	*P. sojae*	Psojae-nad9-R	TTAAATCTATCTTTGTATTCATATATCAA				
	*P. sansomeana*	Psan-nad9-R	TTAGTAGTTAGTACTAATATAACAAAAATATAATA				
*atp9*		Atp9-P	TTGCTTTATTYTGTTTAATGATGGCWTTY (T-FAM) T [THF] A (T-BHQ1) YTTATTTGCTTTTT [3′-C3SPACER]				
*Ty3/Gypsy* retroelement	*C. truncatum*	Pso12-F	CAGGTTTTCAGCGATCTCATCCAAGTG	60	282	qPCR	[171]
		Pso6-R	CACATTGCGGAAAAGGAGGTGATTGCT				
		Pso-P5	FAM-TGCCGACTGCGAGGTCAGCAACCACTTCAA-IBFQ				

^a^ Specificity contradicted by [161,162]. ^b^ Incubation for the LAMP assay. Incubation for 60 min. ^c^ Incubation for the LAMP assay. Incubation for 80 min. ^d^ Incubation for the RPA assay. Incubation for 29 min; details see [169].

**Table 13 jof-09-00587-t013:** Primers and corresponding assays for diagnosis of *Pythium* spp. on soybean.

Target Gene	Target Species (Specificity)	Primer/Probe (Combination)	Sequence (5′-3′)	T_m_ (°C)	Fragment Length (bp)	Assay	Ref.
ITS	*P. ultimum* and HS group	K1	ACGAAGGTTGGTCTGTTG	55		PCR	[175]
		K3	TCTCTACGCAACTAAATGC				
ITS	*P. aphanidermatum*	Pa1	TCCACGTGAACCGTTGAAATC	67 72	210 150	PCR	[176]
	none	ITS2	GCTGCGTTCTTCATCGATGC				
	*P. irregulare*	Pir1	AGCGGCGGGTGCTGTTGCAG				
ITS	*P. aphanidermatum*	AsAPH2B	GCGCGTTGTTCACAATAAATTGC	57 ^a^	163 150	PCR	[177]
	*Pythium*	AsPyF	CTGTTCTTTCCTTGAGGTG	52 ^a^			
	*P. torulosum*	AsTOR6	CGCCTGCCGAAACAGACTAG	59 ^a^			
*Gene encoding a spore cell wall protein*	*P. ultimum*	F3	CAACTGGAAAAGCAAGCGG	64 ^b^		LAMP	[178]
		B3	CCGAAGAACTGTGTCCGC				
		FIP	GAGCCAGACGGGCCAGTATCAAGTTACAGTGGCGTTGTCA				
		BIP	TCTCTGTTGCTCGACTGGAGGGTTCCACCTCCTGTAAGACCT				
		F-Loop	GCTTGCTCCAGTACGAATGC				

^a^ Calculated melting temperatures. The general *Pythium* primer AsPyF can be combined either with AsAPH2B or AsTOR6. ^b^ Incubation for the LAMP assay. Incubation for 60 min.

**Table 14 jof-09-00587-t014:** Primers for amplification of genes useful for phylogenetic analysis and establishing detection assays (in Ascomycetes).

Target	Primer	Sequence (5′-3′)	T_m_ (°C)	Fragment Length (bp)	Reference
ITS	ITS1-F	CTTGGTCATTTAGAGGAAGTAA	54	600	[181]
	ITS4	TCCTCCGCTTATTGATATGC			[182]
*TEF1*	EF1-782F	CATCGAGAAGTTCGAGAAGG	58	350	[183]
	EF1-986R	TACTTGAAGGAACCCTTACC			
*TUB*	Bt-2a	GGTAACCAAATCGGTGCTGCTTTC	60	500	[180]
	Bt-2b	ACCCTCAGTGTAGTGACCCTTGGC			
*CAL*	CAL-228F	GAGTTCAAGGAGGCCTTCTCCC	55	500	[183]
	CAL-737R	CATCTTTCTGGCCATCATGG			
*HIS*	H3-1a	ACTAAGCAGACCGCCCGCAGG	58	450	[180]
	H3-1b	GCGGGCGAGCTGGATGTCCTT			
*ACT*	ACT-512F	ATGTGCAAGGCCGGTTTCGC	61	300	[183]
	ACT-783R	TACGAGTCCTTCTGGCCCAT			

**Table 15 jof-09-00587-t015:** Targets and primers for soybean as reference for quantification relative to soybean and other internal controls.

Target	Primer	Sequence (5′-3′)	T_m_ (°C)	Fragment Length (bp)	Reference
*cox1*	FMPI2b	GCGTGGACCTGGAATGACTA	57		[166]
	FMPI3b	AGGTTGTATTAAAGTTTCGATCG			
	Plant-IC probe	CalFluorRed610-CTTTTATTATCACTTCCGGTACTGGCAGG-BHQ-2			
*cox1*	Cox1-IPC-F	CATGCGTGGACCTGGAATGACTATGCATAGA	39 ^a^		[168]
	Cox1-IPC-R	GGTTGTATTAAAGTTTCGATCGGTTAATAACA			
	Cox1-IPC-P	GGTCCGTTCTAGTGACAGCATTCCYACTTTTATTA [TAM-dT] C [THF] C [BHQ2-dT] YCCGGTACTGGC [3′-C3SPACER]			
*GAPDH*	GmG-14F	CATCGGAGGGAAGTATGAAAGG			[187]
	GmG-14R	GTACAATGCATGATGGTGGC			
	GmG-14HEX	HEX-TTTGTGGGTGACAACAGGTGATGG-IBFQ			
HHIC	HHIC-Fwd	CTAGGACGAG AACTCCCACA T		111	[187]
	HHIC-Rev	CAATCAGCGG GTGTTTCA			
	HHIC-HEX	HEX-TCGGTGTTGA TGTTTGCCAT GGT-IBFQ			
	HHIC ^b^	CACGCCTAGG ACGAGAACTC CCACATCGAG CTTGACGCAA ACGACCACGC CAGGACCATG GCAAACATCA ACACCGAGCG CAACGCCTTG TGCTGAAACA CCCGCTGATT G ^b^			

^a^ Incubation for the RPA assay. Incubation for 29 min; for details see [169]. ^b^ Sequence of the artificial internal control target HHIC (Haudenshield and Hartman internal control).

## Data Availability

Not applicable.

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
