# Peer review of "Diagnosis of Soybean Diseases Caused by Fungal and Oomycete Pathogens: Existing Methods and New Developments"

_jof, 2023, doi:10.3390/jof9050587_

Round 1
Reviewer 1 Report
1. The study was written very scientifically and I congratulate the authors. I hope it will be a highly cited compilation.
2. Soybean production amounts should be given for the whole world, not for the Americas.
3. Literature information for Figure 1 should be re-examined and written.
4. There is a high similarity in different parts and it is necessary to reduce the plagiarism rate in these parts.
5. 37. literature should be cited in the article.
6. 101. Do you need literature? ("compendium" 5th edition is more developed than 4th edition and it is not clear why this source is cited.)
7. My recommendations are given on the attached article.

Reviewer 2 Report
The manuscript is somehow very long. You may consider adjusting the references. Please refer to the text for minor suggestions.

Excellent.
Reviewer 3 Report
The study presented by Hosseini et al. and entitled “Diagnosis of Soybean Diseases caused by Fungal Pathogens: Existing Methods and New Developments” is a very good review, that I found very interesting, useful and clearly written. This article explores the existing methods and the newest developments on diagnosis of soybean diseases caused by fungi and oomycetes. The authors clearly synthesis the exiting information, therefore I feel that the manuscript could be very useful, and a good reference for upcoming studies on the area.
I have only one minor concern; on the title of the manuscript and on line 94 the authors refer to “2. Common fungal soybean pathogens and molecular assays to detect them”, nonetheless in this section the authors are including Phythopthora and Pythium which are, as pointed out by the authors, oomycetes. That is why I suggest changing the name of the section to “Common soybean pathogens” or including fungi and oomycete.
Other than this observation of being careful of not mixing fungi and oomycetes I found the review very acceptable.
Furthermore, I found the conclusions, references, tables and figures very clear and appropriate.
Reviewer 4 Report
The paper provides a description of major fungal pathogens of soybean and of molecular methods by which they can be identified.
Section 2 of the paper provides a useful compilation of PCR diagnostics for genera and species of fungal pathogens including targets, primers, probes and references to their sources. As such this is likely to be a useful resource for soybean pathologists either for adoption of exiting methods, or as a start point for assisting development of additional ones. I have just a few minor specific comments about this text:
Line 131 refers to D. aspalathi, which does not seem to be mentioned in Table 2; D. sojae is also referenced on this line (and elsewhere in text) but apparently referred to as D. soyae in Table 2?
Lines 170 – 175 – do the results of Botelho et al [35] indicate that the primer pair SSFWD/SSREV should be used with caution as being not always reliable? Some conclusion about functionality of the pair would be useful.
At line 181 – by definition all valid genera are ‘monophyletic’; stating this here is therefore either redundant, or providing an indication that other genera reviewed may not share this validity? Hoping for the former, I suggest this word be removed and the sentence slightly revised.
Section 3 is a very general overview and explication of the use and value of molecular technologies in fungal pathogen diagnosis. While written in a satisfactory manner this text provides little advance over similar descriptions that are widely available in textbooks; I would question whether this material is appropriate for publication in this journal. I suggest this section should at least be considerably condensed, but also consider that the manuscript would not suffer greatly from its deletion.
Generally of good quality
